



# Atmospheric Dynamics Reduce Mid-latitude Heatwave Frequency under Idealized Climate Change Forcing

Wolfgang Wicker[1], Emmanuele Russo[2, 3], and Daniela I. V. Domeisen[1, 2]

[1]Université de Lausanne, Lausanne, Switzerland
[2]ETH Zurich, Zurich, Switzerland
[3]European Center for Medium-Range Weather Forecasts, Bonn, Germany

**Correspondence:** Wolfgang Wicker (wolfgang.wicker@unil.ch)

**Abstract.**

Recent decades have seen a global increase in hot temperature extremes, yet the role of changes in the atmospheric circulation in driving this trend remains unclear. To better understand how atmospheric dynamics control extreme weather, we explore a mechanism that relates mid-latitude heatwave frequency to the storm track position in a suite of idealized model experiments

with the dry dynamical core of the ICON model. The underlying relationship between the zonal phase speed of synoptic-scale waves, the latitude of the storm track, and the strength of the eddy-driven jet is assessed through spectral analysis of upper-tropospheric meridional wind. By comparing our experiments to reanalysis data, we find evidence that observed trends in the Southern Hemisphere circulation have contributed towards reducing the persistence of austral mid-latitude hot temperature extremes. This mechanism may also be relevant for the future evolution of extreme events in the Northern Hemisphere, where

we see the joint influence of Arctic Amplification and the expansion of the tropics.

## 1 Introduction

Persistent heat poses a severe hazard to human health (e.g., Ebi et al., 2021). The observed increase in the frequency and intensity of hot temperature extremes in most regions of the world (e.g., Seneviratne et al., 2021; Russo and Domeisen, 2023), therefore, causes great concern and calls for a better understanding of the different drivers of this change. A range of physical

processes has likely contributed to the increase in hot extreme weather in response to anthropogenic climate change (Domeisen et al., 2023; Barriopedro et al., 2023). Foremost, there is the greenhouse effect, which is responsible for the general warming trend and contributes to extreme events (Schär et al., 2004; Fischer and Knutti, 2015). The general warming trend is also linked to an increase in the average saturation vapor pressure (O'Gorman and Muller, 2010), which has various implications for temperature extremes. In a warmer climate, the feedback between low soil moisture and anomalously high air temperature

(Miralles et al., 2014; Hauser et al., 2016) is becoming more effective because moisture deficits can build faster (Dai et al., 2018). Furthermore, as more moisture becomes available for cloud condensation (Joos et al., 2023), upstream latent heat release can contribute more effectively to block formation and a stable stratification that is characteristic of extreme heat near the surface (Zschenderlein et al., 2019; Papritz and Röthlisberger, 2023). These processes and their impact on weather extremes are studied using complex earth system models (e.g., Wehrli et al., 2019).





In addition to the above thermodynamic effects, it has been hypothesized that changes in the atmospheric circulation might contribute to the tendency in extreme events, most prominently in the context of Arctic amplification (Francis and Vavrus, 2012; Cohen et al., 2014; Coumou et al., 2015; Pfleiderer et al., 2019). As Arctic temperatures are rising twice to four times as fast as the global mean (Rantanen et al., 2022; Screen and Simmonds, 2010a, b), we are expecting to see a reduced equator-to-pole temperature gradient near the surface and a weakened westerly jet stream at height (Harvey et al., 2020). For assessing the

relation between Arctic amplification and temperature extremes, we can investigate the strength and the size of the perturbations to the zonally averaged circulation. The sign of the expected waviness response is, however, metric-dependent (Geen et al., 2023). Geometry based metrics predict an increase in waviness and, related to potentially more persistent jet meanders, more extreme mid-latitude weather (Francis and Vavrus, 2012; Cohen et al., 2014). Intensity or kinetic energy based metrics, on the other hand, predict a reduction in waviness; but based on the hypothesis that a weakened circulation will render thermodynamic

feedbacks more effective, the consequence of reduced waviness is again an increase in warm temperature extremes (Coumou et al., 2015; Pfleiderer et al., 2019).

While the near-surface temperature gradient is reducing due to the sea ice loss and changing cloud radiative properties in the Arctic, an increased latent heat release in the tropics is strengthening the upper-tropospheric temperature gradient (Shaw et al., 2016). The circulation response is described as a poleward expansion of the Hadley cells (Lu et al., 2007; Grise et al.,

2019; Grise and Davis, 2020), which is associated with a poleward shift of the extratropical storm tracks and strengthened mid-latitude jet streams (Butler et al., 2010; Chang et al., 2012; Barnes and Polvani, 2013). In the Southern Hemisphere, the greenhouse gas-related signal of tropical upper-tropospheric warming is supported historically by the polar stratospheric cooling due to anthropogenic emissions of ozone-depleting substances (Shindell and Schmidt, 2004; Arblaster and Meehl, 2006; Thompson et al., 2011; Arblaster et al., 2011; Watt-Meyer et al., 2019). Consequentially, a significant positive trend in

the Southern Annular Mode (SAM) is being reported for more than twenty years (Thompson et al., 2000; Marshall, 2003). In the Northern Hemisphere, on the other hand, slowly emerging trends are often regional and strongly dependent on season (Shaw et al., 2024). While reanalysis data points towards a dominance of the tropical warming over Arctic amplification (Woollings et al., 2023), open questions about the strength of certain feedbacks remain (Screen et al., 2022).

The statistical significance of the aforementioned circulation trends is expected to increase in the coming years (Shaw et al.,

2024). Meanwhile, we seek to improve our understanding of how temperature extremes respond to circulation changes by exploring a mechanism that relates the frequency in hot temperature extremes to the spectral properties of large-scale atmospheric waves. Specifically, we focus on heatwaves, defined by a minimum number of consecutive hot days at a given location (e.g., Perkins and Alexander, 2013), and link them to the zonal phase speed of upper-tropospheric disturbances (Fragkoulidis and Wirth, 2020). The phase speed of upper-tropospheric disturbances, or Rossby waves, is used here as a persistence metric

for dynamically forced subsidence and temperature advection (Wicker et al., 2024). Climate models project a decrease in the dominant zonal wavenumber in eddy kinetic energy spectra (Chemke and Ming, 2020) and an increase in phase speed in eddy momentum flux spectra (Chen and Held, 2007) in response to anthropogenic climate change. However, we still lack a full understanding of these projections.





For the first part of this study (Sec. 3,) we follow the modeling setup by (Butler et al., 2010) and perform a suite of experiments that produce a wide range of wave characteristics in a dry dynamical model (see Section 2.1) to shed new light on the relation between heatwaves and waviness. Heatwaves in a dry dynamical model are driven primarily by adiabatic warming due to subsidence and warm air advection (Jiménez-Esteve and Domeisen, 2022), whereas cloud radiative effects and thermodynamic feedbacks are explicitly not simulated. These idealized experiments are therefore well suited to isolate the role of atmospheric dynamics. In the second part of this study (Sec. 4), we compare the idealized modeling results to reanalysis data for the Southern Hemisphere. It is not straightforward to distinguish between radiative effects, thermodynamic feedbacks, and the influence of circulation changes in observed trends from reanalysis data. However, we do find evidence that the poleward shift of the Southern Hemisphere storm track and jet stream has influenced heatwave frequency. Finally, we discuss to what extent the mechanism discovered in the idealized model is relevant in the real world.

## 2 Data and Methods

### 2.1 Idealized experiments

Our model experiments are based on the Held and Suarez (1994) configuration of the ICOsahedral Nonhydrostatic atmospheric model (ICON version 2.6.4; Zängl et al., 2015; Giorgetta et al., 2018; Russo et al., 2025) and run for 50 years of 360 days each with six-hourly output after discarding one year of spin-up. Specifically, we use the R2B4 grid configuration, corresponding to a horizontal resolution of approximately 160 km, and we use 47 vertical levels. In the reference simulation, the atmospheric flow is forced by relaxation to an equilibrium temperature profile and damped by linear friction in the lower troposphere as well as horizontal diffusion of temperature and momentum (Held and Suarez, 1994). Note, in particular, that the model forcing is zonally symmetric and features no seasonal or diurnal cycle.

Following the approach by (Butler et al., 2010), we conduct a range of sensitivity experiments where we can impose circulation changes by forcing the temperature balance with an additional localized, but zonally symmetric heat source (see Appendix A for details). Mathematically, this is equivalent to adjusting the equilibrium temperature profile towards which the circulation is relaxed. First we compare the reference simulation to two selected sensitivity experiments (i.e., exp9 and exp4, Table A1) with either a tropical upper-tropospheric heat source or Arctic surface intensified heat source that produce a 4°N poleward or equatorward storm track shift, respectively (Fig. 1d). To assess potential non-linear effects, we then compare ten additional experiments with combined tropical and Arctic heat sources or sinks. Finally, we compare the model results to ERA5 reanalysis data (Hersbach et al., 2020) for the satellite era from 1979 to 2022. For convenience, upper-tropospheric meridional wind from ERA5 is regridded to a horizontal resolution of 2°x2° and daily-maximum 2 m temperature is regridded to a resolution of 0.5°x0.5°.





## 2.2 Diagnostics

Throughout this study, we define heatwaves as events of at least three consecutive hot days. As our numerical model has no
seasonal cycle, diurnal cycle, or topography, we can define hot days in the model based on a grid point-dependent but otherwise
fixed 90th percentile threshold for daily mean temperature at 1000 hPa. For each experiment, the temperature threshold is set
separately. By definition, each model grid point experiences on average 36 hot days per year. The frequency and duration of
heatwaves are, therefore, solely a function of hot day persistence. In ERA5 data, we define hot days based on a seasonally
varying 90th percentile threshold of daily maximum 2m temperature for a 31-day multi-year rolling window (see, e.g., Russo
and Domeisen, 2023; Brunner and Voigt, 2024). The number of hot days per year at any grid point varies significantly between
the first and the second half of the time series since we do not detrend the ERA5 temperature data.

Our main waviness diagnostic is vertically integrated kinetic energy for 10-day high-pass filtered horizontal wind anomalies
obtained by convolution with a 30-day Lanczos kernel. The maximum of zonal-mean eddy kinetic energy can be referred to
as the mid-latitude storm track (e.g., Shaw et al., 2016), not to be confused with extratropical cyclone tracks from tracking
algorithms of pressure minima (e.g., Sprenger et al., 2017). The spectral properties of Rossby waves are assessed based on
power spectra of meridional wind anomalies at 250 hPa in coordinates of zonal wavenumber and phase speed following
the methodology of Randel and Held (1991). For the idealized model experiments presented in Section 3 we compute the
climatological mean for the entire time series of 1285 individual spectra for 14-day tapered windows. The choice of the
window length is a trade-off between temporal and phase speed resolution (Wicker et al., 2024). The spectra for ERA5 data
presented in Section 4 are computed as the climatological mean of two times 572 spectra, where meridional wind anomalies
are computed with respect to a seasonally varying climatology of 14-day multi-year windows. The wave packet diagnostic in
Figures 4 and 8 is based on the Hilbert transform of composite-mean temperature anomalies that are zonally filtered in Fourier
space for zonal wavenumbers 3-15 (Zimin et al., 2003).

## 3 Modeling results

### 3.1 Storm track latitude and phase speed

The first step in this study is to compare our model simulations to the benchmark by Butler et al. (2010). The zonally symmetric
forcing(Held and Suarez, 1994) generates a potential temperature profile in our reference simulation that approximates equinox
conditions (Fig. 1a). Zonal-mean potential temperature is symmetric around the equator, increases from the surface to the model
top, and features a pronounced meridional gradient in the mid-latitudes. Butler et al. (2011) advocate a simple eddy-diffusive
model to understand circulation changes based on changes in baroclinicity, an expression for the slope between surfaces of
constant potential temperature and surfaces of constant pressure. The sensitivity experiment with tropical forcing shows a
zonally symmetric warming that is largely confined to the tropical upper troposphere (Fig. 1b), enhancing both the meridional
potential temperature gradient in the mid-latitudes and vertical stratification in the tropics and subtropics. The response to the
tropical forcing is, therefore, a dipole of reduced baroclinicity in the subtropics and enhanced baroclinicity further poleward





(Butler et al., 2011). The sensitivity experiment with Arctic forcing shows a surface-intensified warming at the boreal high latitudes reducing stratification and the meridional gradient (Fig. 1c). Mid-latitude eddy heat fluxes are tightly linked to these changes in baroclinicity. The eddy heat flux response to tropical forcing (black contours in Fig. 1b), in particular, mirrors the dipole response of baroclinicity and indicates a poleward shift of the mid-latitude heat flux maximum. The response to Arctic forcing, on the other hand, shows an equatorward shift but also a clear weakening of the heat flux maximum (black contours in
Fig. 1c). These results obtained with the ICON model generally agree with the benchmark from the Colorado State University general circulation model (Butler et al., 2010). To achieve a quantitative agreement, however, we have reduced the amplitude of the tropical forcing, increased the amplitude of the Arctic forcing, and modify its vertical profile (see Appendix A).

     A weakening of the mid-latitude circulation in response to Arctic amplification has been broadly discussed in the context of temperature extremes (Coumou et al., 2015; Pfleiderer et al., 2019). The strength of the circulation can be measured by eddy
kinetic energy or zonal-mean zonal wind. The eddy kinetic energy response to the different forcings (Fig. 1d) resembles the response in the eddy heat flux with a poleward shift in response to tropical warming and an equatorward shift combined with a weakening for the case of Arctic warming. A sensitivity experiment with combined forcing shows a weakening in maximum eddy kinetic energy but no meridional shift (red line in Fig. 1d) indicating a, to large extent, linear circulation response. The zonal wind response to thermal forcing is strongly dependent on altitude (not shown). Here, we use the mass-weighted vertical
average of zonal-mean zonal wind (Fig. 1e) that is determined mostly by the lower troposphere as a proxy for the eddy-driven jet that arises from the stirring of the mid-latitude potential vorticity contrast (Kidston and Vallis, 2012). The location of the eddy-driven jet is closely linked to the mid-latitude storm track. In particular, we find that the maximum of the vertically averaged zonal wind shifts poleward or equatorward by 5°N in response to tropical or Arctic warming, respectively (Fig. 1e). In addition, we find that the poleward shifted jet is strengthened by about 0.9 m/s while the equatorward shifted jet is weakened
by 0.7 m/s. This finding confirms a previously reported relationship between the latitude and the speed of the eddy-driven jet (Kidston and Vallis, 2012). The strength of the eddies, on the other hand, appears less important for the strength of the eddy-driven jet, as indicated by the experiment with the combined forcing (red line in Fig. 1e) where the magnitude of eddy kinetic energy is reduced while the location of its maximum and the strength of the jet are largely unchanged. In the following, we advocate for using a shift of the storm track or jet latitude as a metric for quantifying the response of mid-latitude dynamics to
anthropogenic climate change.

     We start by assessing the zonal phase speed of synoptic-scale disturbances in the upper troposphere. The Hayashi spectra in Figure 2a-c depict the phase speed spectrum of meridional wind variance as a function of zonal wavenumber. In general, the spectra show high variance for wavenumbers 5 to 8 and an increasingly eastward phase speed for increasing zonal wavenumber from less than 0 m/s to roughly 10 m/s. The spectral information can be summarized by the centroid (the "center of mass") of
the spectrum indicated by black markers in Figure 2a-c. For the reference simulation, the centroid lies at a phase speed of 4.3 m/s and zonal wavenumber 6.3 (black circle). Tropical and Arctic warming cause statistically highly significant, dipole-shaped differences in power spectral density (Fig. 2b, c) in the respective sensitivity experiments. The centroid is shifted to a higher phase speed of 5.2 m/s and a lower wavenumber of 5.9 in response to tropical warming (black cross in Fig. 2b), whereas the centroid is shifted to a lower phase speed of 3.1 m/s and higher wavenumber of 6.6 in the case of Arctic warming (black cross







**Figure 1.** (a) Zonal-mean potential temperature $\bar{\theta}$ (shading) and mean meridional eddy heat flux $\overline{v'\theta'}$ (black contours with spacing of 3 Kms$^{-1}$, negative values dashed) in the reference simulation and (b)-(c) the response in $\bar{\theta}$ and $\overline{v'\theta'}$ (contour spacing of 1 Kms$^{-1}$) in the sensitivity experiments with a tropical and a polar heat source, respectively. Lines of (d) vertically integrated eddy kinetic energy and (e) mass-weighted vertically averaged, zonal-mean zonal wind for the reference simulation, the two experiments with a single forcing, and one experiment with combined tropical and Arctic forcing with a color shading that indicates the 95% confidence interval based on a bootstrapping of annual means.





in Fig. 2c). A baseline for understanding how the phase speed of upper-tropospheric disturbances might be related to the storm track latitude is given by the Rossby wave dispersion relation from linear theory. More specifically, the zonal phase speed $c_p$ of barotropic Rossby waves can be expressed as a function of mean zonal wind $\bar{u}$, the meridional planetary vorticity gradient $\beta$, and the zonal and meridional wavenumber $k$ and $l$.

$$c_p = \bar{u} - \beta/(k^2 + l^2) \tag{1}$$

Both the mean zonal wind $\bar{u}$ and the meridional planetary vorticity gradient $\beta$ are functions of latitude. Therefore, in Figure 2d-e, we plot the centroid of the Hayashi spectra averaged over smaller meridional bands of 5°N width, compared to the broad meridional range of 35°N to 65°N used for the spectra in Figure 2a-c, against latitude. A local maximum in phase speed around the core of the storm track (Fig. 2d) and a rapid decrease in zonal wavenumber towards higher latitudes (Fig. 2e) are clearly visible. Comparing the experiments with tropical and Arctic warming, phase speed is increased or decreased over the
entire range of latitudes as the storm track is shifted poleward or equatorward, respectively. Based on equation (1), this could be explained by a lower $\beta$ at higher latitudes. More plausibly, however, the relationship between phase speed and storm track latitude is explained by the strength of the vertically averaged zonal-mean zonal wind or eddy-driven jet (Fig. 1e). Following the reasoning of Kidston and Vallis (2012), the relationship between latitude and strength of the eddy-driven jet, in turn, is explained by the inverse proportionality between the eddy stirring efficiency and the meridional absolute vorticity gradient $\beta^*$.

## 3.2 Persistent temperature extremes

Next, we assess the response in persistent temperature extremes to the idealized thermal forcing. Heatwave frequency has, according to the definition employed here, an upper bound of on average 36 days/year. In that case, every hot day that exceeds the 90th percentile threshold for near-surface temperature forms part of a heatwave. Our reference simulation shows a heatwave frequency above 30 days/year in the low and high latitudes close to the theoretical maximum (Fig. 3a). In the mid-latitudes,
however, about 5°N poleward of the EKE maximum, heatwave frequency reduces to approximately 14 days/year. There, less than 40 % of the hot days form part of a heatwave with a minimum duration of 3 days. The strongest response in heatwave frequency to the idealized forcing is seen between 45°N and 65°N (Fig. 3a). The sensitivity experiment with Arctic warming shows an increase, whereas the experiment with tropical warming shows a decrease in heatwave frequency. An alternative way to summarize the results is to say that the mid-latitude heatwave frequency minimum shifts in coherence with the EKE
maximum between 43°N and 52°N. The magnitude of minimum heatwave frequency is reduced or increased for a poleward or an equatorward shift of the EKE maximum, respectively.

A mid-latitude minimum, similar to the one for heatwave frequency, is also seen for heatwave duration (Fig. 3b). This mid-latitude minimum shifts meridionally in a similar fashion as the frequency minimum. Its magnitude, however, does not change significantly with values for meridional minimum mean heatwave duration between 3.71 days and 3.77 days. For a better
understanding of the different responses in heatwave frequency and duration, we plot the distribution in hot day persistence as a function of latitude in Figure 3c-d: The distribution of heatwave duration is strongly skewed. While, in exceptional cases,




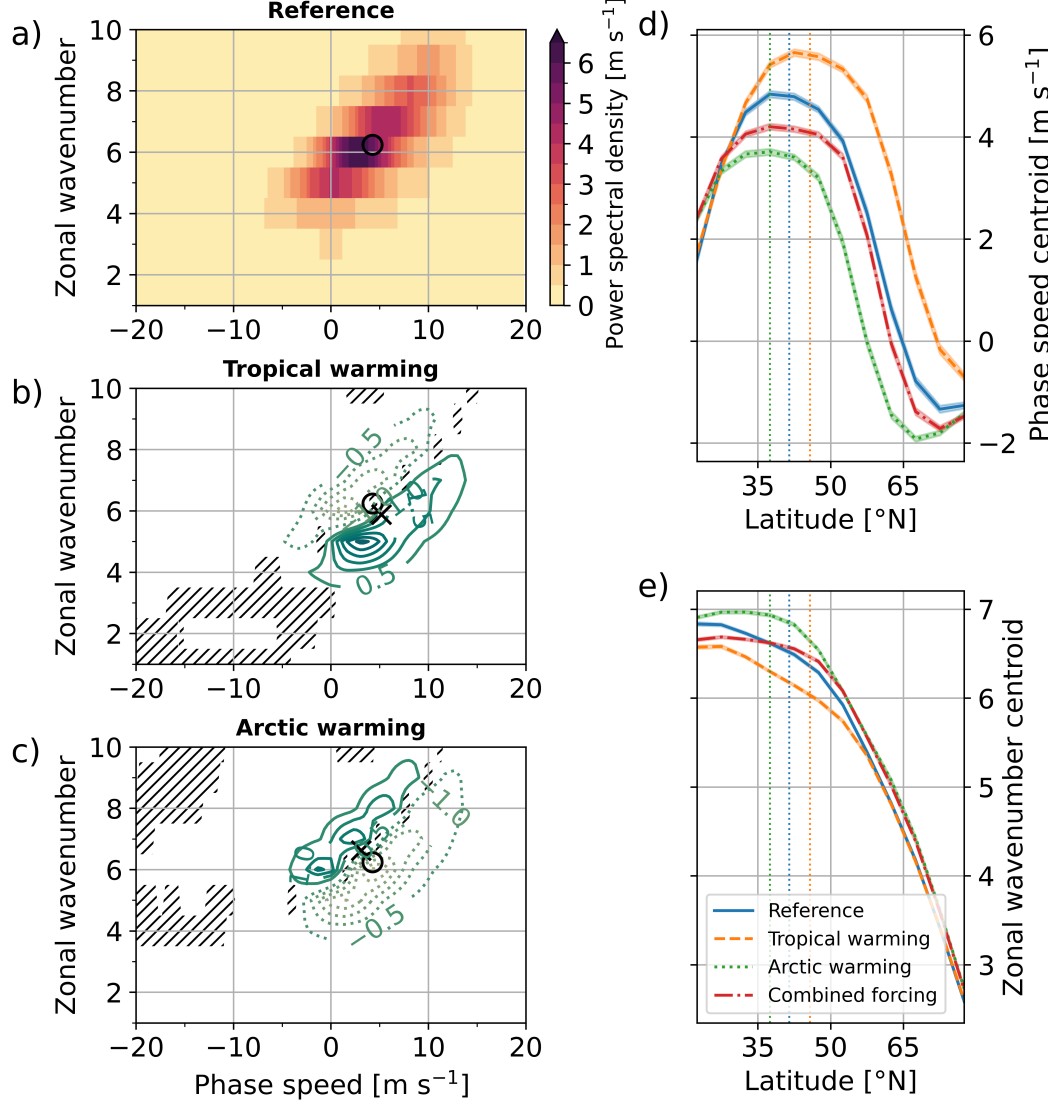

**Figure 2.** Climatological-mean meridional-mean (35°N to 65°N) power spectral density of meridional wind at 250 hPa for (a) the reference simulation and (b)-(c) the difference in the experiments with tropical and Arctic warming, respectively. The circle indicates the centroid of the meridional-mean spectrum for the reference simulation, the 'x' markers indicate the centroid for the respective sensitivity experiment. Hatching indicates where the spectrum of the reference simulation and the sensitivity experiment are not significantly different at the 99.9% confidence level based on a parametric bootstrap of 14-day windows. The centroid of (d) phase speed spectra and (e) zonal wavenumber spectra averaged over smaller 5°N latitude band widths is plotted as a function of central latitude with the 95% confidence interval (shading). The dotted vertical lines in (d)-(e) indicate the location of the maximum eddy kinetic energy.





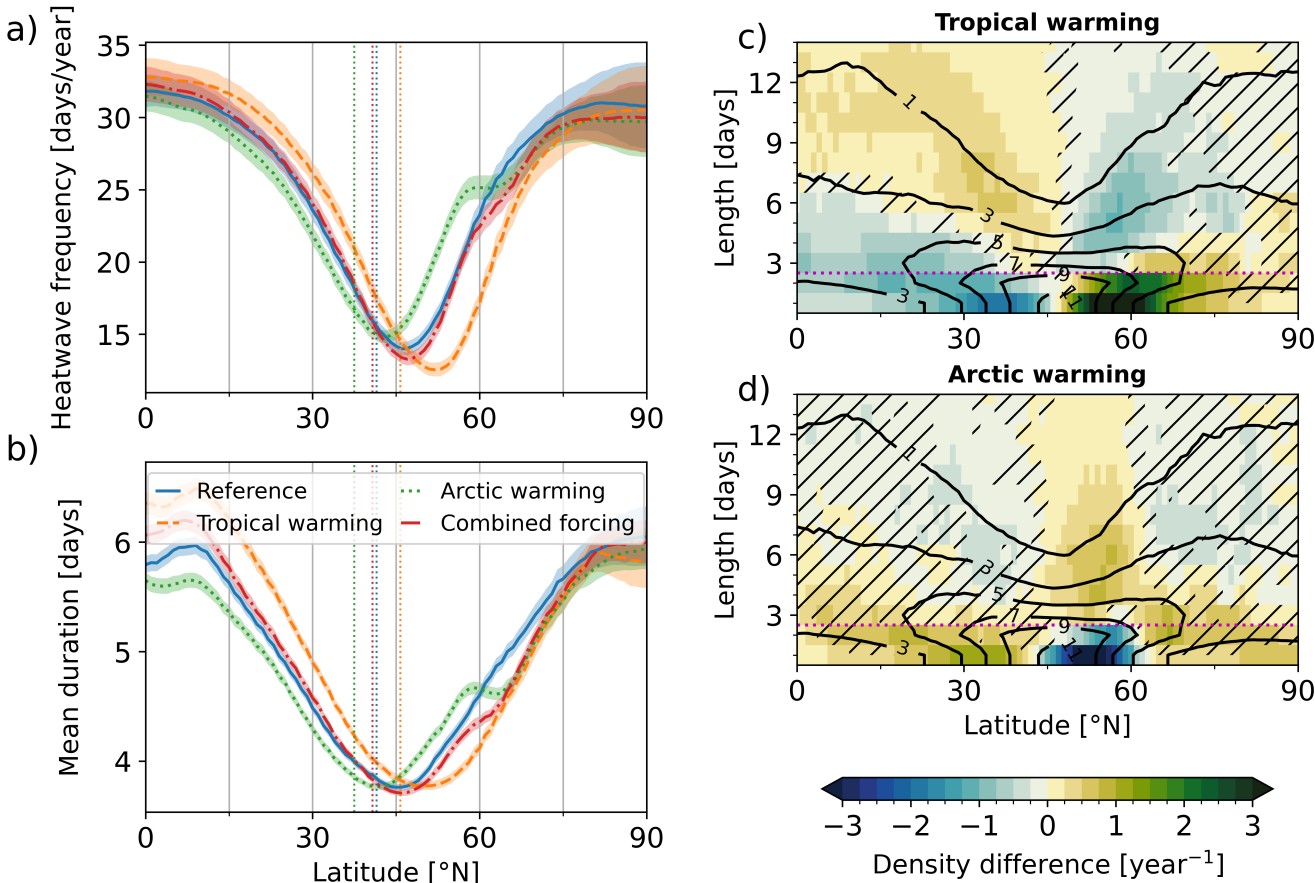

**Figure 3.** Zonal mean of (a) heatwave frequency and (b) mean heatwave duration. The color shading indicates the 95% confidence intervals. The storm track location in terms of the maximum eddy kinetic energy is indicated by dotted vertical lines for the reference simulation and the three sensitivity experiments. Changes in zonal-mean hot day persistence with respect to the reference run as a function of latitude for (c) tropical warming and (d) Arctic warming. Hatching indicates where the hot day density in the respective sensitivity experiment is not significantly different from the reference simulation at the 95% confidence level and black contours indicate hot day density in the reference simulation in units of $\mathrm{year}^{-1}$. Integrating the density difference at a given latitude over length yields 0 $\mathrm{days/year}$, while integrating hot day density for reference simulations yields 36 $\mathrm{days/year}$.

heatwave duration can reach two weeks or more, most mid-latitude heatwaves last only as long as the minimum duration of 3 consecutive days. The lower bound imposed by the definition does not allow mean heatwave duration to vary significantly. Instead a change in mid-latitude hot day persistence imprints strongly on heatwave frequency. This can be seen Figure 3c-d,

where the pivot point between a positive and a negative hot day density response to tropical or Arctic warming lies at a length between 2 and 3 $\mathrm{days}$.





The mid-latitude minima in heatwave frequency and duration point towards a limit or an upper bound imposed on hot day persistence by certain characteristics of the atmospheric circulation. To better understand the driving mechanism for this limit, we compute composite-mean temperature anomalies with respect to the 90th percentile in coordinates of time and longitude relative to a heatwave onset at the latitude of minimum heatwave frequency (Fig. 4a-b). Positive anomalies at the center of the Hovmöller plot indicate the occurrence of hot days. Negative anomalies with respect to the 90th percentile in the upper-left and lower-right quadrant of the plot indicate close-to-climatological conditions. The key feature in these Hovmöller plots is, however, the succession of more and less negative anomalies indicating the eastward group propagation (solid black line) of a wave packet (amplitude in white contours) as time progresses. The closed contours of more or less negative temperature anomalies (blue color shading) represent the cold and warm phases of the wave packet before and after a local amplification that facilitates the occurrence of hot days. Mid-latitude heatwaves in dry dynamical models and in reanalysis data are driven by locally amplified Rossby wave packets (Jiménez-Esteve and Domeisen, 2022; Fragkoulidis et al., 2018). Compared to the wave packet envelope, the eastward phase propagation of individual warm and cold phases, marked by the dotted black lines, happens at a much lower pace. The comparison of the Hovmöller plots of the two experiments with tropical and Arctic warming (Fig. 4c) yields minor differences in the speed of eastward group propagation. The differences in phase propagation and zonal wavenumber (slope and spacing of dotted lines), on the other hand, are very clear and in agreement with the differences shown by the Hayashi spectra of upper-tropospheric meridional wind (Fig. 2). This agreement emphasizes the role played by the upper troposphere for for the persistence of temperature extremes near the surface.

For a closer look at the limit to hot day persistence, we now focus on the positive anomalies at the center of Figure 4a-b where the composite mean exceeds the hot day threshold. One mechanism to limit the duration of a heatwave is the succession by an amplified cold phase of the wave packet. This would be the case for rapid phase propagation when the frequency of the carrier wave does not support a longer heatwave. A second mechanism is the gradual diminution of anomalies as the duration of wave packet amplification is limited by the width of the wave packet and its group velocity. In our model, mid-latitude heatwave duration is limited by the second mechanism: Most heatwaves last for only 3 days, whereas half a period of the carrier wave amounts to approximately 4.5 and 6.5 days. The similarity in group velocity (thin solid contour in Fig. 4c) for both sensitivity experiments can therefore explain the lack of a significant difference in minimum mean heatwave duration (Fig. 3b). Heatwave frequency, however, is limited by phase speed rather than group velocity. This can be seen from the stronger tilt of the hot phase (blue thick solid contour in Fig. 4c) for the case of tropical warming compared to the case of Arctic warming, which increases the longitudinal extent of positive temperature anomalies with respect to the 90th percentile. A faster phase propagation creates additional hot days upstream and downstream of the central longitude. However, there the occurrence of a warm phase is not synchronized with the amplification of the wave packet. The additional hot days upstream and downstream do not occur in a set of consecutive days that is long enough to be classified as a heatwave, but they are preceded or followed by the cold phase of a wave packet. Since the number of hot days is by definition limited, these isolated hot days lead to a reduction of heatwave frequency overall.





**Figure 4.** Composite-mean temperature anomaly at 1000 hPa with respect to the 90th percentile for (a) 17,408 heatwaves at 52°N in the experiment with tropical warming and (b) 23,305 heatwaves at 43°N in the experiment with Arctic warming in days relative to the heatwave onset (y-axis) and longitude relative to the grid point where the heatwave is identified (x-axis). White contours indicate the strength of the wave packet envelope in units of K. Dashed black lines indicate contours of zero phase angle for the carrier wave, while the solid black lines indicate a constant phase angle of the wave packet envelope. The lines of constant phase angle for both experiments shown in (a) and (b) are plotted for comparison in (c) with the solid contour of 0 K composite-mean temperature anomaly.





### 3.3 Position overrules strength of mid-latitude eddies

In the previous Section, we explored a mechanism that relates a mid-latitude minimum in heatwave frequency to the position of the extratropical storm track. The final step in this first part of our study is to evaluate this mechanism over a broader range of circulation changes. Summarizing the mid-latitude heatwave signature by the position (Fig. 5a, b) and the value (Fig. 5c, d) of minimum heatwave frequency, we find a linear behavior over a range of storm track positions from approximately 35°N to 47°N and storm track magnitudes from approximately $460 \, \mathrm{kJ \, m^{-2}}$ to $540 \, \mathrm{kJ \, m^{-2}}$. Two experiments, where the storm track is either shifted far poleward to 51°N by strong tropical warming (exp1) or enhanced to $576 \, \mathrm{kJ \, m^{-2}}$ by Arctic cooling (exp11), form outliers from that linear behavior.

A very high correlation of $R = 0.98$ between the position of the heatwave frequency minimum and that of the storm track maximum confirms the above-reported behavior across all 13 experiments (Fig. 5a). On average, the heatwave frequency minimum is found 5.8°N poleward of the storm track maximum. A significant negative correlation of $R = -0.80$ is found between the minimum value of heatwave frequency and the position of the EKE maximum (Fig. 5c). Specifically, the minimum heatwave frequency is reduced by $-0.25 \pm 0.12$ days per year for every degree of poleward storm track shift. Removing the two positive outliers, the correlation strengthens to $R = -0.93$ and the slope steepens to $-0.35 \pm 0.11$ days per year per degree. Fitting a linear model for the logarithm of minimum frequency yields a 2% reduction per degree of poleward shift. In comparison, the maximum value of EKE energy has no significant predictive power, neither for the position nor for the value of minimum heatwave frequency (Fig. 5b, d). We conclude that, in the absence of thermodynamic feedbacks, the intensity of mid-latitude waves does not contribute to the heatwave frequency response to anthropogenic climate change.

## 4 Clear trends in reanalysis data

Over the past 50 years, the Southern Hemisphere (SH) has experienced a positive trend in the Southern Annular Mode (SAM, e.g., Marshall, 2003) connected to a poleward storm track shift (e.g., Chang et al., 2012). Based on the relationship explored above, we can expect a phase speed increase connected to the observed circulation changes. Indeed, the Hayashi spectra for the SH mid-latitude upper troposphere from ERA5 (Fig. 6a) share the principal characteristics with those from our idealized simulations: a predominantly eastward phase propagation, an increase in phase speed with wavenumber, and a broad peak in power spectral density for synoptic-scale waves. This peak, however, is located at a slightly lower wavenumber and a clearly higher phase speed compared to the ICON reference simulation. In addition, the spectrum for reanalysis is broader than the modeled spectrum, covering a wider range of phase speeds up to almost 20 m/s. Despite these differences between the model and reanalysis, the phase speed trend qualitatively meets the expectation. Comparing the spectra of upper-tropospheric meridional wind variance between the first and the second half of the ERA5 time series (Fig. 6b), the difference in power spectral density forms a dipole similar to the response of the Tropical warming experiment (Fig. 2b). Quantitatively, however, the difference in power spectral density is relatively small: The centroid of the spectrum is shifted from zonal wavenumber 5.61 and a phase speed of 8.46 m/s for the years 1979 to 2000 to zonal wavenumber 5.59 and a phase speed of 8.71 m/s for 2001





**Figure 5.** Position of (a, b) the heatwave frequency minimum and (c, d) the minimum value of heatwave frequency plotted as a function of (a, c) the position of the EKE maximum and (b, d) the maximum value of EKE with a least-square linear model fit for 13 model experiments with different heating profiles. The orange color saturation of the marker edge encodes the strength of tropical heating while the purple color saturation of the marker edge encodes the strength of Arctic heating. The coefficient of determination and the slope of the regression line with its 95% confidence interval is shown for each model fit in the title of the respective subplot and the 95% confidence interval of the response variable is indicated by a grey color shading.





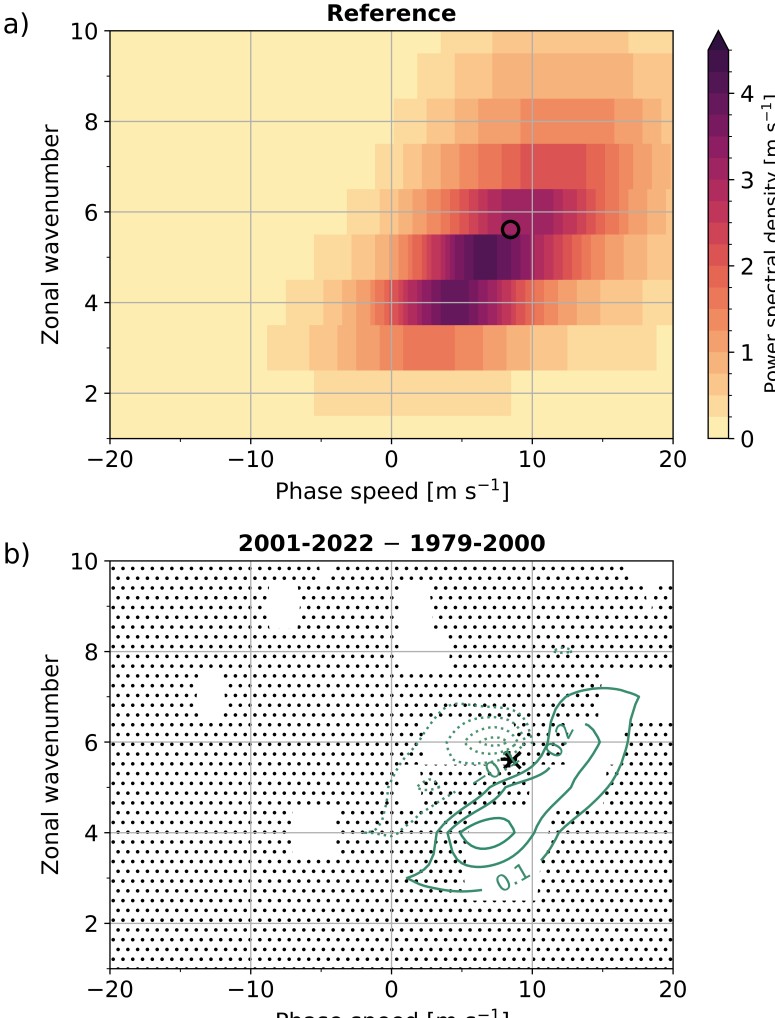

**Figure 6.** (a) Climatological-mean, meridional-mean (35°S to 65°S) power spectral density of meridional wind at 250 hPa and (b) the mean difference between the second and the first half of the ERA5 dataset. Stippling indicates that the spectra are not significantly different at the 95% confidence level. The circular marker indicates the centroid of the spectrum averaged over the entire time series (a). The '+' marker indicates the centroid for the first half of the time series (b), whereas the 'x' marks the centroid for the second half.

to 2022. This increase in phase speed for meridional wind variance is qualitatively in line with trends in eddy flux covariance spectra (Chen and Held, 2007).

Seeing this phase speed increase over time, we naturally wonder how circulation changes have affected heatwave frequency and hot day persistence in the SH mid-latitudes. To address this question, we start by comparing trends in year-round zonal-mean heatwave frequency in ERA5 from 1979 to 2022 between the Northern Hemisphere (NH) extratropics, the tropics, and the SH extratropics (Fig. 7a). Poleward of about 20°N, the NH has experienced a gradual increase in heatwave frequency



from an average of 12 days/year in the 1980s to around 26 days/year in the 2010s. The tropics are marked by pronounced interannual variability where extrema in 1983, 1987, 1998, 2010, 2015-16, and 2019-21 are associated with El Niño events in the tropical Pacific (e.g., McPhaden et al., 2006). The SH, on the other hand, shows a broad mid-latitude minimum in heatwave frequency, much clearer than the NH, and a poleward migration of this SH minimum (Fig. 7a). The pronounced minimum in hot day persistence around 50°S (white contours in Fig. 7b), where more than 70% of hot days occur isolated or as a set of two consecutive days, bears a strong resemblance to the ICON reference simulation.

Given this resemblance between SH heatwave data and our idealized simulations, an evident hypothesis is to attribute the poleward shift of the SH heatwave frequency minimum to storm track processes and the trend of the SAM. We note, however, that the poleward shift of the frequency minimum is very strong and exceeds the expected signal from the idealized model. Comparing the first to the second half of the time series, the minimum in heatwave frequency is shifted poleward by 9°S to 12°S. Furthermore, the difference in SH hot day persistence between the first and second half (Fig. 7b) does not resemble the quadrupole in the experiment with tropical warming (Fig. 3c). Instead, we see a dipole with an increase in hot days equatorward of 60°S and a decrease poleward of that latitude. Opposing trends in Southern Ocean sea surface temperatures to both sides of the Antarctic Circumpolar Current (Armour et al., 2016) have likely contributed to this dipole in the number of hot days and the strong poleward shift of the heatwave frequency minimum. We do, however, notice a small change in persistence as the increase in SH mid-latitude hot days is strongest for a low persistence from 1 to 3 days, whereas the reduction affects sets of consecutive hot days with a length of 2 to 6 days.

Another useful characteristic for comparing SH heatwave data to our idealized model is the magnitude of minimum heatwave frequency. In the SH, the zonal-mean value increases from 10 days/year for the first half of the ERA5 time series to 12 days/year during the second half. This increase does not meet the expectation from our model for a poleward shifted heatwave frequency minimum and raises the question of to what extent the mechanism explored in Section 3 is at play in the SH. For further insight, we investigate composite-mean temperature anomalies relative to the grid point and the onset of a heatwave at the location of the respective heatwave frequency minimum in Figure 8. Similar to the composite mean from model data (Fig. 4), the downstream development of a wave packet is clearly visible. The eastward propagation of the wave envelope (Fig. 8a-b, black solid lines) is faster than the propagation of individual warm and cold phases (dashed lines). Positive composite-mean anomalies at the central longitude last for the minimum heatwave duration of 3 days, as is the case in the model. For the composite mean from reanalysis data, however, the wave packet is composed of only three separate warm phases and two cold phases.

The tilt and the spacing of individual warm and cold phases express the phase speed and the wavelength of the heatwave-generating wave packet. Comparing the first (orange dashed lines in Fig. 8c) and the second half (blue dashed lines in Fig. 8c) of the reanalysis data at the latitude of their respective minimum heatwave frequency, we find an increase in both phase speed and wavelength indicated by a stronger tilt and a larger spacing, consistent with the change in upper-tropospheric phase speed spectra (Fig. 6b). Importantly, we also find a more strongly tilted hot phase in the center of the Hovmöller plot for the second half of the reanalysis data (Fig. 8d, thick solid contours). In the idealized model, isolated hot days upstream and downstream of the heatwave, equivalent to a more strongly tilted hot phase, cause a reduction in heatwave frequency as phase speed increases.

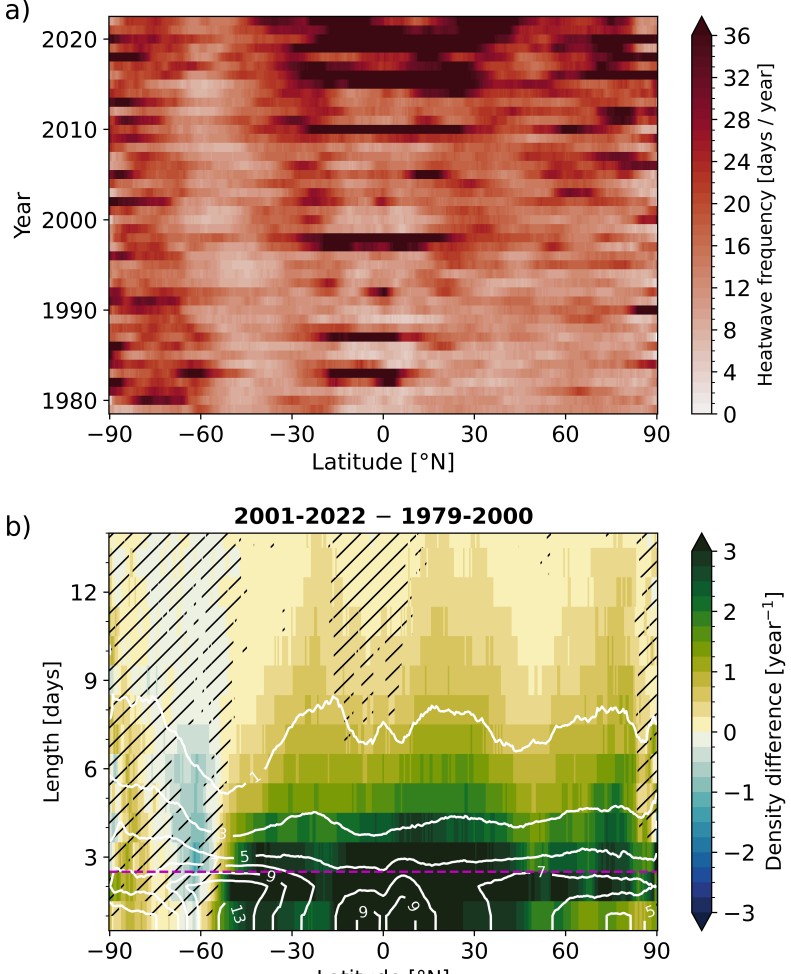

**Figure 7.** (a) Zonal-mean heatwave frequency in the ERA5 dataset and (b) changes in zonal-mean hot day persistence from the first to the second half of the ERA5 time series. Hatching indicates where the hot day density is not significantly different between the two halves of the time series at a 95% confidence level. White contours indicate the hot day density averaged over the entire time series in units of year$^{-1}$.




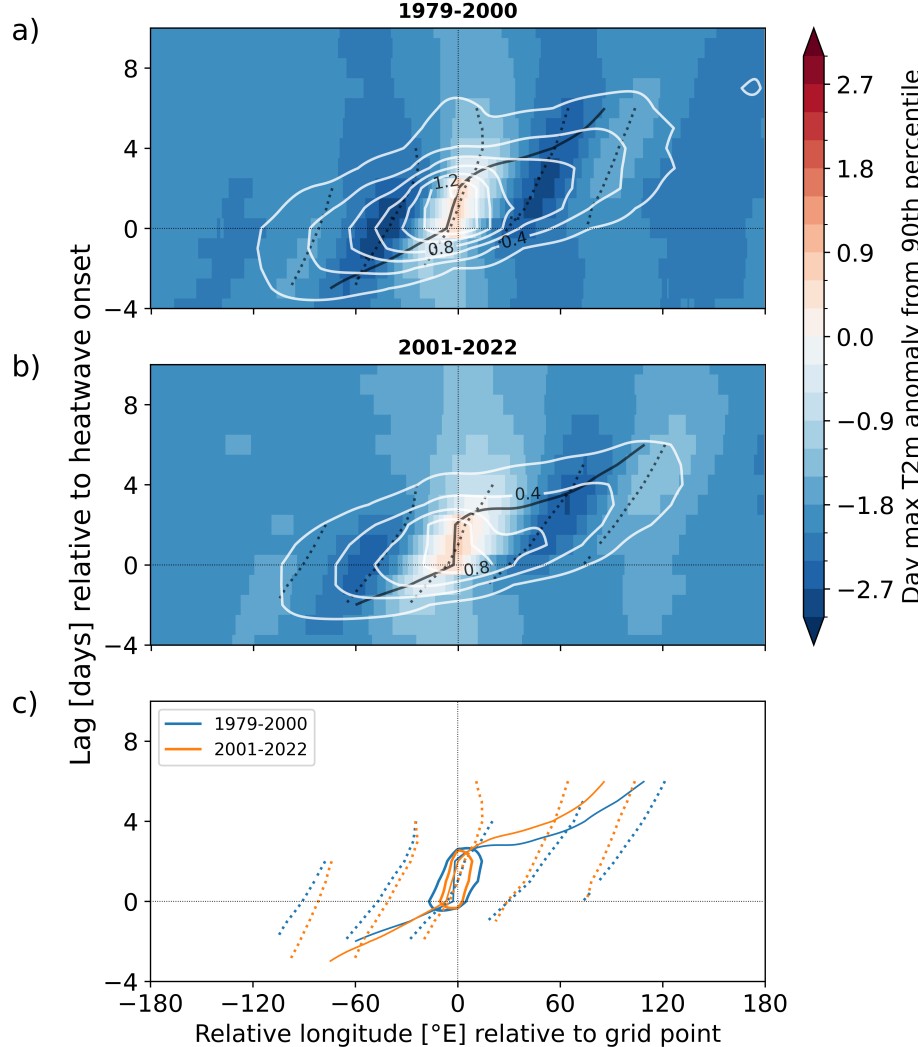

**Figure 8.** As in Figure 4 but with composite-mean 2 m temperature anomalies with respect to the 90th percentile at the location of minimum heatwave frequency at (a) 46.5°S for 1979-2000 and (b) 58.5°S for 2001-2022.

The Hovmöller plot of composite-mean temperature anomalies is therefore an indication that the contribution by atmospheric dynamics to the change in heatwave frequency is also negative for the Southern Hemisphere mid-latitudes during the satellite era. The increase in heatwave frequency that is noted above should, therefore, be attributed to processes other than atmospheric dry dynamics.





## 5 Conclusions

In this study, we explore a mechanism through which atmospheric circulation changes contribute to the heatwave frequency response to anthropogenic climate change. Following up on a volume of literature about the influence of Arctic amplification on mid-latitude weather (Francis and Vavrus, 2012; Cohen et al., 2014; Coumou et al., 2015; Pfleiderer et al., 2019), we assess, in particular, the relationship between heatwave frequency and changes in waviness, here measured by eddy kinetic energy (EKE). Similar to the setup of Butler et al. (2010), we conduct a series of idealized experiments with a dry dynamical general circulation model subject to thermal forcing that approximates the effect of Arctic amplification and the expansion of the Tropics. In these experiments, we identify a pronounced heatwave frequency minimum approximately 6° poleward of the mid-latitude EKE maximum, our proxy for the extratropical storm track. The primary model response to the idealized forcing in terms of temperature extremes is a meridional shift of the heatwave frequency minimum. The magnitude of EKE does not show a significant relationship with heatwave frequency or duration. In contrast, we find the latitudinal position of the extratropical storm track to be a strong predictor for heatwave characteristics. The experiments with tropical warming show a coherent poleward shift of the storm track maximum and the heatwave frequency minimum, whereas the experiments with Arctic warming show a coherent equatorward shift. The second important model response is a 2% reduction of minimum heatwave frequency per degree of poleward storm track shift. We find a significant negative correlation of $R = -0.80$ between the minimum heatwave frequency and the storm track position over the broad range of different levels in waviness in our 13 experiments.

The heatwave frequency reduction in response to a poleward storm track shift can be explained by a significant relationship between the zonal phase speed and the latitudinal position of synoptic-scale waves. Specifically, a poleward-shifted storm track by 4° is linked to an increase in phase speed by approximately 1 m/s and a reduction of the dominant zonal wavenumber by 0.4 . We hypothesize that this relationship is caused by the previously known connection between the strength and the location of the eddy-driven jet (Kidston and Vallis, 2012). A composite analysis of Hovmöller plots shows that the spectral changes in terms of phase speed and wavenumber apply not only for synoptic-scale waves in a climatological sense but also for the heatwave-generating Rossby wave packet. The Hovmöller plots illustrate how an increase in phase speed creates isolated hot days upstream and downstream of a heatwave. These additional hot days, however, do not occur in a set long enough to be classified as a heatwave themselves, and thereby reduce heatwave frequency or hot day persistence overall.

Finally, we assess the relevance of the above-presented mechanism for observed circulation changes in reanalysis data. Over the satellite era from 1979 to present, the Southern Hemisphere (SH) has experienced a significant trend in the Southern Annular mode and a poleward storm track shift (e.g., Marshall, 2003; Chang et al., 2012). In agreement with an analysis of eddy momentum flux spectra (Chen and Held, 2007), we find a phase speed increase of upper-tropospheric meridional wind variance in reanalysis data that resembles the model response to tropical warming. While we cannot distinguish between dynamic and thermodynamic effects in reanalysis data as cleanly as in the idealized model, we do present evidence for the influence of storm track dynamics on SH heatwave frequency. In resemblance to the dry dynamical model, the SH mid-latitudes display a pronounced heatwave frequency minimum and a marked poleward migration of this minimum over the past 50 years. Trends in



Southern Ocean sea surface temperatures have likely left their mark on hot day frequency (Armour et al., 2016). A composite analysis of Hovmöller plots, however, confirms that the change in phase speed of synoptic-scale waves has reduced hot day persistence in the SH mid-latitudes. Moreover, we expect that the above-presented mechanism will be relevant for the future evolution of temperature extremes in the Northern hemisphere as well, given the strong linear relationship between minimum

340 heatwave frequency and storm track position across our experiments with combined tropical and Arctic heating.

*Code and data availability.* The ERA5 reanalysis data can be downloaded from the Copernicus Climate Data Store. The ICON model output is available from authors upon request. A code repository for the data analysis is available under a BSD license from https://doi.org/10.5281/zenodo.15018094.

## Appendix A:  Heating profiles

345 The profiles of the tropical and Arctic heat source $Q_T$ and $Q_A$ are zonally symmetric functions of latitude $\phi$ and the vertical coordinate $\sigma$. Note that the Arctic heating profile is slightly modified compared to Butler et al. (2010). The different experiments are defined by the respective heating amplitudes $q_T$ and $q_A$.

$$Q_T(\phi,\sigma) = q_T \exp\left[-\left(\frac{\phi^2}{0.32} + \frac{(\sigma-0.3)^2}{0.0242}\right)\right] \tag{A1}$$

$$Q_A(\phi,\sigma) = q_A \cos^{15}(\phi-1.57)e^{2\sigma-2} \tag{A2}$$



**Table A1.** The heating amplitudes for the reference and twelve sensitivity experiments.

| Experiment ID | $q_T$ [K/day] | $q_A$ [K/day] |
|:---:|:---:|:---:|
| ref | 0 | 0 |
| exp1 | 0.5 | 0 |
| exp2 | 0 | 0.5 |
| exp3 | 0.25 | 0 |
| exp4 | 0 | 1.0 |
| exp8 | 0.2 | 1.0 |
| exp9 | 0.2 | 0 |
| exp10 | 0 | 1.5 |
| exp11 | 0 | -1.0 |
| exp12 | -0.2 | 0 |
| exp13 | 0.25 | 0.5 |
| exp14 | 0.25 | 1.0 |
| exp15 | 0.5 | 1.0 |

350 *Author contributions.* WW and DD have jointly designed the study. ER has performed the model runs. WW preformed the data analysis, made the figures, and wrote the original draft. All authors contributed to editing the manuscript.

*Competing interests.* One of the co-authors is a member of the editorial board of *Weather and Climate Dynamics*. The authors have no other competing interests to declare.

*Acknowledgements.* The authors would like to thank Andries de Vries and Hilla Afargan-Gerstman for many fruitful discussions. This
355 project has received funding from the European Research Council (ERC) under the European Union's Horizon 2020 research and innovation programme (grant agreement No. 847456). This work was supported by a grant from the Swiss National Supercomputing Centre (CSCS) under project ID s1144.



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
