# Peer review of "A Poleward Storm Track Shift Reduces Mid-Latitude Heatwave Frequency: Insights from an Idealized Atmospheric Model"

_EGUsphere, 2025_

## Author Comment (AC1)

**Final Author Comments**

egusphere-2025-1197

Wolfgang Wicker, Emmanuele Russo, Daniela Domeisen

First of all, we would like to thank the reviewers for their insight and for the time they took to review our work. Their comments pointed out weaknesses in our argument about a change in the persistence of Southern Hemisphere temperature extremes. In particular, the Major Comment 3 by Reviewer 1 and the Major Concern 2 by Reviewer 2 motivated substantial revisions. In the original manuscript, we concluded that the observed phase speed increase applies to heatwave-generating weather systems. This conclusion was based on a comparison of Hovmöller composites between the first and the second half of the ERA5 time series at different latitudes. The revised Figure 8 with a comparison at 50°S shows that this is not the case. In the revised manuscript, we conclude that the observed phase speed increase does not influence the persistence of Southern Hemisphere mid-latitude temperature extremes. While the revised conclusion reduces the relevance of our idealized experiments for the Southern hemisphere, we would like to point out that our experiments might still be relevant for the future evolution of the Northern Hemisphere. In particular, we want to note that the mean phase speed spectrum of the model is much closer to the mean spectrum of the Northern hemisphere than to the spectrum of the Southern hemisphere. We have revised the manuscript accordingly.

Please find our detailed responses to the reviewers' comments below. Additions and changes are marked in blue font color in the revised manuscript. Sentences or paragraphs that have been removed are replaced by brackets [...]. All line indications in red refer to the new (annotated) version of the manuscript.

**Reviewer 1**

**Major comments:**

1. *A comparison with the observed SH storm track trends is attempted, but even though the main contributor to these trends is polar stratospheric cooling via ozone depletion, no polar stratospheric cooling experiment is performed. Though it's true that the tropical warming experiment also results in a poleward shifted jet, the dynamics involved are somewhat different (e.g., see Butler et al. 2010), and there is little discussion of this in the text. In addition, I find the ERA5 SH analysis confusing because as shown in, e.g., Banerjee et al. 2020, since 2000 there has been some indication of ozone recovery in the atmospheric circulation, so that if the early period is subtracted from the later as is done in Figure 6 and 7b, it should reflect not the ozone depletion period but rather the ozone recovery signal (e.g., a weakening and equatorward shift of the jet, see Banejee et al. 2020 Fig 2c,f,i). Though this signal has weakened with more recent years added on (e.g. Shaw et al. 2024), it still is not clear why these two periods were selected, rather than just the ozone depletion period (e.g., year 2000 minus 1979) or perhaps the first period minus the second (opposite of what is shown). Visually at least, it also appears in Figure 7a that the noted shift in SH heatwave frequency minima was larger between 1979 and 2000 compared to 2000-2020, which would line up with this slowdown in the SH poleward jet shift. But then interpreting Fig 6b and 7b is not quite clear since it's a difference between these two periods.*

    **Answer:** Thank you for this comment. The comparison of our idealized model with Southern Hemisphere trends is motivated by the phase speed increase associated with the positive trend in the SAM. Based on your comment, we realized that plotting the linear trend in Figures 6b and 7b is more intuitive than plotting the difference between the second and the first half of the ERA5. We include a reference to Banerjee et al. (2020) in l. 50 to acknowledge the pause in trends due to ozone depletion. However, we would like to point out that the new plot in Figure 6c indicates a continued increase in phase speed during the 2000s

and the 2010s. We include a note on this finding in ll. 280 f. of the revised manuscript.

2. *The background information/motivation in lines 25-48 and 303-305 could be improved/reorganized. It seems odd, for example, to delve into Arctic amplification and details of waviness metrics, and not mention the counteracting effect of upper tropical warming and polar stratospheric cooling until the second paragraph, which feels like an after-thought. Also, the phrasing of lines 28-30 is not exactly true in terms of the "weakened westerly jet stream" part; if only Arctic amplification were occurring, this may be valid, but the fact of the matter is, there are other effects and as such what we've actually seen is weak poleward shifts of the jet (Woollings et al. 2023). There is also lack of discussion of seasonality of these effects, which seems important to mention; for example, while there seems to be more solid evidence of a sea ice effect on the circulation in the NH summer, this effect does not seem detectable in the winter. The ozone depletion signal is also going to be primarily important in austral spring/summer. Overall, I suggest trying to rewrite so that the contributing "forcings" are all discussed first, and then explaining how these are expected to (or already have) driven trends in the NH and SH separately.*

**Answer:** Thank you for this suggestion. We realized that the original title and abstract of the manuscript might have been misleading about the scope of our manuscript. We do not aim to assess the shape of the forcing that leads to a certain change in the storm track or in terms of heatwave characteristics. Instead, we explore the relationship between the storm track position, the phase speed of synoptic-scale waves, and mid-latitude heatwave frequency. Note, in particular, that the forcing of the idealized model is tuned in shape and amplitude to produce a certain storm track shift (ll. 136 ff.) instead of aiming to replicate the specific influence of greenhouse gas forcing. We have updated the title and abstract to better convey the scope of the manuscript and have adapted the introduction to the new content of the revised manuscript.

3. *While the authors acknowledge this issue, it is difficult to disentangle the thermodynamic effects (and also, oceanic effects) in the ERA5 reanalysis. The authors do not try to detrend the ERA5 temperature data, but I do wonder if there would be some way to try to isolate the dynamic changes from the thermodynamic changes better. Perhaps by removing the global-mean from each year and grid box? I also wonder whether the minimum in heatwave frequency shows up so nicely in the SH in 7a because this is where the Southern Ocean takes up much of the heat, and its colocation with the storm track is thus coincidental.*

   **Answer:** Thank you for this comment. As outlined above, we changed our interpretation of the Southern Hemisphere trends in response to Major Concern 2 raised by Reviewer 1. During the revision of this manuscript, we experimented with multiple ways of detrending the time series, and we realized that the poleward shift of the heatwave frequency minimum is a consequence of the mean temperature trend and the associated trend in hot day frequency. Please note our discussion of this issue in ll. 292 ff.

**Minor Comments**

*Line 31: This is the first mention of "waviness" which may confuse unfamiliar readers. Could an explanation of waviness be linked somewhere in line 29 to the weakened jet stream? (as a general note, maybe the in-depth description of waviness metrics lines 31-36 should instead be moved to the Methods section?)* **Answer:** Thank you for your comment. We modified the section in ll. 35 ff. to be clearer.

*Line 34-35: I don't understand what is meant by "render thermodynamic feedbacks more effective"; could more explanation be provided?* **Answer:** Thank you for the comment. We rephrased in l. 40.

*Line 42-43: "supported historically"- it's unclear what this is supposed to mean. Do you mean, the trends due to ozone depletion are in the same direction as those due to greenhouse gases, and so they are additive? Or do you mean, the modelled poleward trends due to tropical upper tropospheric warming are in the same direction as observed trends due to ozone depletion?* **Answer:** Thank you for the comment. We mean that storm

track trends due to ozone depletion and greenhouse gases are in the same direction. We rephrased in ll. 46 ff.

*Line 45: See Major comment, but "more than twenty years" is a strange period to discuss here (especially when the references themselves are from 20+ years ago- but surely those references were looking at SAM trends over the decades before they were published?). It should be mentioned that these trends have weakened in the last twenty years due to ozone recovery, and the Banerjee et al. 2020 paper should be cited.* **Answer:** Thank you for the comment. We rephrased in ll. 49 ff. and cite Banerjee et al. (2020).

*Line 46-48: This jumps back to the NH; see Major comment about reordering/reorganizing these paragraphs.* **Answer:** Thank you for the comment. Please see our response to Major Comment 2.

*Line 47: Another reference here could be Lee et al. 2019 (https://www.nature.com/articles/s41586-019-1465-z)* **Answer:** Thank you for this suggestion. The work by Lee et al. (2019) discusses the relationship between wind shear and the meridional temperature gradient at different altitudes. In contrast, our results highlight the influence of the vertically averaged zonal wind. Therefore, we believe that a reference to Lee et al. (2019) might be confusing here.

*Line 49: Can you explain why (e.g., because the signal will start to exceed the noise)?* **Answer:** This sentence was a reference to Shaw et al. (2024). We replaced it in l. 57.

*Line 56-57 (also lines 257-258): are these results in idealized models, or realistic? For both hemispheres, or just one or the other?* **Answer:** Thank you. We now specified in l. 64.

*Line 72: Can you comment about whether things like the SAM and associated jet shift in response to a forcing can be captured well by this configuration (e.g., Gerber et al. 2008, Chan and Plumb 2009)? Are there implications for your results?* **Answer:** The resolution of our model configuration is sufficient to represent synoptic-scale variability accurately. Jiménez-Esteve and Domeisen (2022) and Russo et al. (2025) use the same model configuration to study the role of atmospheric dynamics for heatwaves. But, after all, the idealized model is not meant to capture every feature of the atmospheric circulation.

*Line 81: It would be nice to either add to this table or have a separate table that summarizes/quantifies some of the key changes in jet shift, phase speed, etc across all*

*the experiments.* **Answer:** Thank you for this suggestion. We summarized the circulation changes across all experiments in the new supplement (Table S1).

*Line 97 (also, lines 306, 317): Is EKE really the best metric for "waviness", per se? I think of waviness as contemporaneous regions of high and low pressure around a longitude circle, but EKE is highest in the storm tracks. In Geen et al. (2023), this is not even mentioned as a waviness metric. Alternatively, maybe "waviness" isn't exactly the right word for what you're evaluating here?* **Answer:** Thank for this comment. We rephrased in l. 107 of the revised manuscript.

*Line 134: This sensitivity to altitude is shown in Butler et al. (2010) so could refer to that here* **Answer:** Thank you. We adopted your suggestion in l. 144.

*Line 139-141: could the fact that the Tropical heating exp (poleward shifted jet) has a larger magnitude strengthening than the Arctic amplification (equatorward shifted jet) weakening magnitude be due in part to the stronger magnitude imposed heating in the Tropical heating case compared to the Arctic amplification case?* **Answer:** The eddy-driven jet strength (mass-weighted vertical average) is not directly related to the forcing profiles. According to Kidston and Vallis (2012), the jet strength is determined by the strength of the eddy stirring and the stirring efficiency.

*Line 143-145: It's confusing whether this line is a conclusion of the line before it, or a new thought to be demonstrated in the analysis below.* **Answer:** Thank you for your comment. We removed the sentence from l. 154.

*Line 164-165: zonal wavenumber on the other hand only strongly changes in the mid-latitudes in the sensitivity experiments (Fig 2e)* **Answer:** Thank you. We include a note on the influence of thermal forcing on the zonal wavenumber in l. 181.

*Line 178: perhaps should be mentioned that the opposite is true from ~25-40N* **Answer:** Thank you for the suggestion. We focus on the latitude range where we see the strongest signal poleward of the storm track maximum

*Figure 2: would be useful to mention in caption or on figure that "positive" values are "eastward"* **Answer:** Thank you for the suggestion. The definition of positive as eastward is standard and implied by the statement in l. 158.

*Lines 223-224: So, are the results here truly dynamically relevant, or do they arise in part or as a side effect of the threshold definition chosen here? Also, I think a few more lines could be added re: the question posed at the top of this page, which is, is there a limit on*

*hot day persistence due to certain characteristics of the atmospheric circulation, and why? I think lines 210-225 are trying to get at that, but it would be nice to re-summarize what this means. For example, lines 226 says "In the previous section, we explored a mechanism..." but it was not exactly clear to me what the mechanism was.* **Answer:** Thank you for the comment. It is not clear to us what "truly dynamically relevant" means. The link between heatwave frequency and phase speed is of course linked to the definition of a heatwave as a persistent temperature extreme. Our results should be robust against choosing a slightly different percentile threshold or minimum duration, as long as heatwaves do not become a very rare event. A typical Rossby wave packet in the idealized model can hardly support a 6-day heatwave.

*Line 239-240: This quantification is useful, but one thing that is unclear is if it's location specific (e.g., is it only true at the heatwave minimum latitude?). Or does it more generally mean, a poleward shifted jet means less frequent mid-latitude heatwaves overall?* **Answer:** Thank you for your comment. We include a plot that confirms the linear relationship for meridionally averaged (45°N-65°N) heatwave frequency in Figure S3d. Please see the reference to Figure S3d in ll. 260 f.

*Line 241-242: parts of this sentence are seemingly contradictory ("in the absence of thermodynamic feedbacks" and "in response to anthropogenic climate change"- which is going to be dominated by thermodynamics). Also, don't you really mean in response to latitudinal shifts in the jet (which could be caused by climate change or ozone depletion or even natural forcings)?* **Answer:** Thank you for the comment. We removed the sentence in l. 262 to increase clarity.

*Line 246: where is this shown? Fig 2d?* **Answer:** Yes. This expectation is based on Figures 2c and 2d. Indeed, the new Figure 6c shows strong resemblance of Figure 2d.

*Line 260: describe based on idealized results in previous section what you expect to see in terms of heatwave frequency and hot day persistence* **Answer:** Thank you for this suggestion. We prefer to avoid repetition of the results based on the idealized model in Section 4.

*Line 282: by "does not meet the expectation from our model"- what would the expectation be? Could this be better quantified, as it was for the model results?* **Answer:** Thank you for the comment. We extensively revised this part of the manuscript

accordingly to our new interpretation of the data. The equivalent to the sentence referenced in this comment is now found in ll. 302 ff.

*Line 300-301: where is it noted above? Fig 7c? At the heatwave density minimum or generally? This statement also seems to contradict lines 66-67, which argues that there is a role. Overall, could the conclusions in lines 298-301 be better explained? I'm not sure I follow how the Hovmoller plot means that the dynamics reduce heatwave frequency.*
**Answer:** Thank you for the comment. We rephrased the end of Section 4 in ll. 309 ff.

*Line 329-332: It should again be mentioned the main reason for this poleward shift, which is only partly increasing greenhouse gases and mostly ozone depletion- which was not an experiment tested here and which might have different effects on the jet than tropical warming (since the dynamics involved are different).* **Answer:** Thank you for the suggestion. We revised the end of Section 5 according to our new interpretation of the data.

**Technical Edits**

*Line 8: "hot temperature extremes" is repetitive, can just say "hot extremes" (same in line 35 and elsewhere)* **Answer:** Thank you. We rephrased in l. 2, and l. 19.

*Line 16, 17: instead of "general" ◊ "global"* **Answer:** Thank you for the suggestion. We believe that there is a difference between "general" and "global" and that "general" is suited better in ll. 22 f.

*Line 22: "block formation" – not clear what is meant here, though I think you mean atmospheric blocking. Could change to "atmospheric high-pressure 'blocking' formation"* **Answer:** Thank you. We rephrased in l. 28.

*Line 29: "at height" – specify at which height* **Answer:** Thank you. "At height" is thought to be equivalent to "above the surface" here.

*Line 37: specify "NH" in front of "near-surface temperature gradient"* **Answer:** Thank you. We believe that the NH is evident from the sub clause "sea ice loss and changing cloud radiative properties in the Arctic".

*Line 39: here, specify "circulation response to tropical upper tropospheric warming"* **Answer:** Thank you. We add the word "associated" in l. 44.

*Line 47: here, specify that you mean "on the atmospheric jet streams" or some equivalent in terms of the dominance of tropical warming over Arctic amplification* **Answer:** Thank you for the suggestion.

*Line 59: move comma outside the parenthesis* **Answer:** Thank you. We changed l. 68 accordingly.

*Line 112: put a space between "forcing" and parenthesis* **Answer:** Thank you. We changed l. 122 accordingly.

*Line 127: change "modify" to "modified"* **Answer:** Thank you. We changed l. 137 accordingly.

*Line 132: is this exp 8?* **Answer:** Yes. We changed l. 142 accordingly.

*Line 133: change to "indicating, to a large extent, a linear circulation…"* **Answer:** Thank you for this suggestion. We believe that there is a semantic difference depending on where the "a" is placed.

*Line 138: here 5N is mentioned but in line 82 is what said the shift was 4N* **Answer**: The correspondence between the location of the EKE maximum and the jet maximum is not perfect. The EKE maximum is shifted by 4°N, while the jet is shifted by 5°N. Note that Figures 1d and 1e indicate uncertainty.

*Line 204: "lower" -> "slower"* **Answer:** Thank you for this suggestion.

*Line 208: remove second "for"* **Answer:** Thank you. We rephrase the sentence in ll. 225 f.

*Line 242: use of "waves" here is a bit confusing since it could be atmospheric waves or heat waves- suggest using "eddies" (to also match section title)* **Answer:** Thank you. We removed the sentence in l. 262.

*Line 272-273: is this the delta latitude, or the actual latitude? I think the former but it's not clear from the wording* **Answer:** It is not clear to use what delta latitude means. We believe that the formulation "shifted poleward by 9°S" is correct.

*Line 296: "Fig 8d" should be "Fig 8c"* **Answer:** Thank you for this comment.

*Line 319: specify "mid-latitude" heatwave frequency reduction?* **Answer:** Thank you for this suggestion.

**Reviewer 2**

**Major Concerns**

1. *It is unclear whether the shift in the position of SH storm tracks between the first and second halves of the ERA5 reanalysis is driven by SAM, tropical expansion, or a combination of both. The response of the power spectral density in Figure 6b resembles the result from the idealized tropical warming experiment (Figure 2b), which also produces a poleward shift in storm tracks. However, the underlying causes of tropical expansion and the positive phase of SAM are different. I agree with Referee #1 that without polar stratospheric ozone depletion experiments (or alternative experiments with stratospheric cooling) [e.g., 1], it is hard to establish a causal link between observed changes in heatwave frequencies and SAM. In addition, given that the jet shift associated with SAM is most pronounced in austral summer (i.e., December to February), I wonder why the authors analyze the entire year rather than focusing exclusively on that season.*

   **Answer:** Thank you for raising this concern. As explained in the response to Major Comment 2 of Reviewer 1, we do not aim at establishing a causal link between a specific forcing (tropical upper-tropospheric warming due to greenhouse gases or polar stratospheric cooling due to ozone depletion) and certain circulation changes (i.e., the storm track shift). Instead, we explore the link between the storm track position, the eddy-driven jet, and the zonal phase speed with its implications for temperature extremes. Hence, the forcings are used as a way to produce different storm track positions that will then allow us to draw conclusions about dynamical connections to heatwaves. Figure S3 in the new supplement illustrates the strong linear relationship between storm track position and the strength of the eddy-driven jet or the zonal phase speed for the entire set of our 13 experiments. This supplementary Figure is referenced in the manuscript in ll. 246 ff. Note, in particular, that tropical cooling (exp12) leads to an equatorward storm track shift and a phase speed reduction, while Arctic cooling (exp 11) leads to a poleward storm track shift and a phase speed increase. We conclude that the strength of the eddy-driven jet and the phase speed of synoptic-scale waves depend directly

on the storm track position and less on the shape of the forcing. Based on your comments, we agree that it would be interesting to include experiments with stratospheric cooling. However, we do not believe that such experiments are necessary to support the main conclusions of the paper. A discussion of the seasonal cycle of the link between heatwaves and the atmospheric circulation lies beyond the scope of our manuscript.

2. *The observed responses of near-surface temperature and SH storm tracks are also closely linked to ocean dynamics [e.g., 2, 3]. For example, the stronger and poleward-shift westerlies due to ozone depletion before 2000 can increase northward Ekman drift and upwelling south of the ACC. The northward Ekman drift leads to northward expansion of the sea ice and a colder SST around 60° As enhanced Ekman drift also pumps up warm waters from below the mixed layer, a slow warming trend is expected to reverse the initial cooling near 60°S over decades [3]. The above mechanisms might be able to explain why there is a more notable poleward shift in SH heatwave frequency before 2000 compared to 2000 to 2020 period.*

   **Answer:** Thank you for raising this concern. In the original manuscript, we referred to the study by Armour et al. (2016). Based on your concern, we realized that this reference is not sufficient and extended the discussion of Southern Ocean sea surface temperatures in ll. 292 ff. of the revised manuscript. Most importantly, we came to the conclusion that the phase speed increase over time has not influenced the persistence of hot extremes (l. 298). As discussed in ll. 309 ff., we attribute this discrepancy between our idealized experiments and the Southern Hemisphere to differences in the climatological-mean spectrum. In contrast to the model, a typical Rossby wave packet in reanalysis is not persistent enough to sustain a 3-day heatwave. In the Southern Hemisphere, the duration of heatwaves is limited by the eastward phase propagation of the upper-tropospheric Rossby wave. The climatological change in phase speed can, therefore, not affect heatwave-generating weather systems. We rephrased our conclusions accordingly in ll. 12 ff., ll. 298 ff., and ll. 348 ff.

**Specific Comments:**

*L5-7: Consider clearly mentioning in the Abstract that the idealized forcing experiments represent Arctic warming and tropical expansion.* **Answer:** Thank you for this comment. We revised the Abstract extensively. In particular, we now better specify the nature of the model forcing in ll. 5 f. We also state the role of ozone-depletion in the Southern Hemisphere in ll. 10 f.

*L8: I am not sure if there is strong evidence supporting the impact of SH circulation on midlatitude hot extremes. In addition, the term "austral" is only mentioned here, but the ERA5 analysis appears to cover the entire year.* **Answer:** Thank you for this comment. In ll. 11 f. of the revised abstract, we justify the comparison of the model with the Southern Hemisphere by the reminiscence of the mid-latitude heatwave frequency minimum. In our response to Major Comment 3 by Reviewer 1 and to Major Concern 2 by Reviewer 2, we elaborate on the evidence for reduced persistence of SH temperature extremes. We removed the term "austral" from the Abstract.

*L9-10: This statement is oversimplified, given that the future response in the NH hot extremes could also be influenced by topography, land-sea contrast, atmosphere-ocean coupling and interactions with sea ice, in addition to Arctic amplification and tropical expansion.* **Answer:** Thank you for this comment. In our opinion, the concluding remark of the abstract in ll. 15 f. is justified by our experiment design and the literature introduced in ll. 31-46 and ll. 53-56. Admittedly, our idealized model cannot represent the influence of every element of the climate system. However, the idealized nature is clearly stated in the revised title.

*L31-34: I am not sure why the authors mention "waviness" here. The rest of the manuscript is discussing the latitude and the strength of the eddy-driven jet instead of jet waviness.* **Answer:** Thank you for your comment. We modified the section in ll. 35 ff following the Reviewer's comment. Indeed, the rest of our manuscript discusses the strength and the latitude of the storm track. But we think it is necessary to introduce the "waviness" discussion in the context of weather extremes.

*L45: SAM also has a strong seasonality, but the authors did not mention it when analyzing the ERA5 data.* **Answer:** Thank you for your comment. A discussion of the seasonal cycle

of the link between heatwaves and the atmospheric circulation lies beyond the scope of our manuscript.

*L45-47: "Does 'the slowly emerging trends' refer to the shift in extratropical storm tracks, or to something else?"* **Answer:** The expression refers to circulation trends in general and is a quote from the reference in l. 54 (Shaw et al., 2024).

*L49: The authors could elaborate more why the statistical significance of the circulation trends will increase.* **Answer:** Thank you for the comment. We replaced the sentence in l. 57.

*L59: (Sec. 3,) -> (Sec. 3),* **Answer:** Thank you for spotting this typo in l. 68.

*L60: (see Section 2.1,) -> (Section 2.1)* **Answer:** Thank you for spotting this typo in l. 69.

*L81: Suggest providing a few sentences to clarify the reasoning for selectin exp4 and exp9.* **Answer:** As stated in the manuscript, the experiments were selected because they both produce a 4° storm track shift.

*L86-87: Why are different horizontal resolutions used for winds and near-surface temperature in the ERA5 analysis.* **Answer:** For the analysis of synoptic-scale variability, the 2°x2° resolution is sufficient and chosen to reduce computational costs. We don't think that our results are sensitive to this choice, and we refrain from justifying it in the manuscript. For temperature extremes, on the other hand, we use data close to the native model resolution underlying ERA5.

*L90: Since the "heatwave frequency minimum" is quite important for the later analysis, suggest providing a clear definition earlier in the Methods section.* **Answer:** Thank you for this suggestion. We do not think that stating the simple definition of a meridional minimum improves the flow of the manuscript.

*L93-94: This sentence "In ERA5 data…rolling window" is very confusing. Just to clarify, do you mean that for each calendar day, you first consider a 31-day window centered on that day and gather temperature data from multiple years to compute the 90$^{th}$ percentile? In that case, the 90$^{th}$ percentile temperature would change with time.* **Answer:** Yes, indeed, the temperature threshold for defining hot days varies with the seasonal cycle (i.e., one value for each day of the year). This is common practice, and we provide two references in ll. 103f.

*L96: Could you clarify why the ERA5 temperature data were used without detrending?* **Answer:** Because we are interested in the trend and display its effect in Figures 7 and 8.

*L97: I am not sure if "waviness" is the correct word to refer to the vertically integrated EKE. To measure "waviness", you should consider metrics like finite local wave activity [4] or meridional circulation index [5].* **Answer:** We agree with this comment and changed the formulation in l. 106 accordingly.

*L112: forcing(Held and Suarez, 1994) -> forcing (Held and Suarez, 1994)* **Answer:** Thank you for spotting this typo in l. 122.

*L124: The response to Arctic forcing... -> The response of eddy heat flux to Arctic forcing...* **Answer:** Thank you. We implemented this suggestion in ll. 133 f.

*L139-141: I don't know whether the differences in jet strength are related to the different forcing magnitudes and profiles used in exp4 and exp9.* **Answer:** The eddy-driven jet strength (mass-weighted vertical average) is not directly related to the forcing profiles. According to Kidston and Vallis (2012), the jet strength is determined by the strength of the eddy stirring and the stirring efficiency. We show how the storm track latitude (related to the stirring efficiency) is more important that the storm track strength (related to the stirring strength).

*L141-143: I am not sure I fully understand this sentence.* **Answer:** In ll. 150 f., we discuss the latitude of the jet and the storm track with a reference to Kidston and Vallis (2012). In ll. 152 ff. we compare the influence of the strength of the storm track based on experiment exp8.

*L151: statistically highly significant -> statistically significant* **Answer:** Thank you. We adopted your suggestion in l. 161.

*L172: according to the definition employed here -> according to the definition employed in Section 2.2* **Answer:** Thank you. We adopted your suggestion in l. 184.

*L184: A question beyond the main discussion: why does heatwave duration show significant differences in low latitudes but not in high latitudes in Figure 3b?* **Answer:** That is a good question. We don't know to this moment. The present study focuses on the mid-latitudes because that is where we expect the spectral properties of Rossby waves to influence heatwaves. Furthermore, we are cautious to interpret the low and high latitudes where the temperature balance in the model is altered by thermal forcing.

*Figure 3c and 3d: What is the pink dotted line? The figure caption for 3c and 3d is a bit confusing. By saying "Changes in zonal-mean hot day persistence with respect to the reference run…", audience are not expecting the probability density difference plots here.*

**Answer:** The magenta dotted grid line indicates the minimum length to be classified as part of a heatwave. We improved the explanation of Figures 3c-d in ll. 198 ff.

*L193-195: Suggest clarifying in main text which latitude is used for the composite-mean temperature anomalies.* **Answer:** Thank you for the suggestion. We replaced Figure 4 in with a comparison at 50°N. Since this new Figure shows a comparison at a fixed latitude, we see the influence from the relative distance to the storm track maximum (i.e. the amplitude is much higher for Tropical warming experiment)

*L201: In reanalysis data, the midlatitude heatwaves are also influenced by local land-atmospheric feedback, soil moisture, convection and other factors. Actually, the first paragraph of Introduction has already acknowledged that many other processes besides Rossby wave amplification are responsible for heatwaves in the real world.* **Answer:** Thank you for your comment. We removed the sentence in l. 220 and placed a rephrased sentence without reference to reanalysis data in ll. 208 f.

*L221-224: I am uncertain why "3 consecutive days" is chosen as threshold here. Some other heatwave metrics, such as the Heat Wave Duration Index (HWDI), use five consecutive days as the duration to classify a heatwave. Therefore, I suggest adding some justifications why this threshold is chosen.* **Answer:** Thank you for comment. Heatwaves with a length of 5 days or more are rare in the model and particularly in Southern Hemisphere mid-latitudes because the period of a typical Rossby wave packet cannot support a 5-day anticyclone. In this study, we focus on moderate extremes, and 3 consecutive days is a valid choice commonly used in literature (e.g., Russo et al., 2015, Env. Res. Let.). Please also see the modified discussion in ll. 312 ff.

*L230: I don't think there is a linear relationship between heatwave frequency minimum (or position of heatwave frequency minimum) and storm track strength.* **Answer:** The phrasing in ll. 249 f. refers to the set of experiments without the outliers exp1 and exp11.

*L234: On average, the heatwave frequency minimum... -> On average, the latitude of the heatwave frequency minimum...* **Answer:** Thank you. We adopted your suggestion in ll. 254 f.

*L241-242: I am not sure what the "absence of thermodynamic feedback" refers to. The authors could simply say "in an idealized model with Held-Suarez configuration".* **Answer:** Thank you for the comment. We removed the sentence in l. 262.

*L246: Suggest clarifying which regions are expected to experience an increase in phase speed.* **Answer:** Thank you for your suggestion. We do not think that is necessary to specify that the phase speed increase is expected in the SH mid-latitudes.

*L253: Are circulation data in ERA5 detrended?* **Answer:** No, the reanalysis data is not detrended. The power spectra are computed for meridional wind anomalies from the climatological-mean.

*L254: Suggest deleting "however".* **Answer:** Thank you for this suggestion. We modified the sentences in ll. 271 ff. and add the discussion of the new Figure 6c in ll. 275 f.

*L260: I am confused by the word "trends". It seems that Figure 7a is not a trend plot.* **Answer:** The Figure 7a depicts the zonal-mean heatwave frequency from 1979-2022. Trends in that quantity differ across latitudes.

*Figure 7b: What is the pink dotted line?* **Answer:** We removed the magenta grid line from Figure 7b.

*L269-270: I am not sure whether attributing the poleward shift of observed heatwave frequency minimum to the trend of the SAM is correct. Please see major concern (1).* **Answer:** Thank for this comment. We replaced the full paragraph about the persistence of SH temperature extremes from the original manuscript with a new paragraph in ll. 292 ff.

*L271: What is the "expected signal from the idealized model"?* **Answer:** The expected signal is a reduction in minimum heatwave frequency. However, we revised the last paragraph of Section 4 accordingly to our new interpretation of the data.

*L275-277: Could the authors elaborate more how ocean dynamics affect the observed SH midlatitude heatwaves? Please see major concern (2) for detailed comments.* **Answer:** Thank for this comment. We replaced the full paragraph about the persistence of SH temperature extremes from the original manuscript with a new paragraph in ll. 292 ff.

*Figure 8c: the orange line should be 1979-2000 and the blue line should be 2001-2022.* **Answer:** Thank you for spotting this mistake. The new Figure 8a shows the composite-mean over the full time series and the new Figure 8b shows a comparison of the decades 1980-2019.

*L295-296: Should both figures referred to be Figure 8c?* **Answer:** The new Figure 8b shows the comparison of composites at a fixed latitude. We revised the end of Section 4 accordingly.

*L208-300: I am not sure I fully understand this sentence.* **Answer:** Thank you. We revised the end of Section 4.

*L300-301: Where is the increase in heatwave frequency noted? Figure 7a?* **Answer:** We revised the end of Section 4 extensively.

*L306: As mentioned earlier, "waviness" shouldn't be used here.* **Answer:** Thank you for the comment. We rephrased in ll. 320 f.

*L317: What does "over the broad range of different levels in waviness" mean?* **Answer:** We rephrased in l. 332 to "broad range of different storm track characteristics".

*L335-336: The argument of how Southern Ocean SST influences the hot day frequency is weak and non-convincing.* **Answer:** Thank you for the comment. We rephrased in ll. 348 ff.